# Two-Stage Input-Space Image Augmentation and Interpretable Technique for Accurate and Explainable Skin Cancer Diagnosis

**Catur Supriyanto** [1,2,*] ⬥, **Abu Salam** [1,2] ⬥, **Junta Zeniarja** [1,2] ⬥ and **Adi Wijaya** [3] ⬥

1  Faculty of Computer Science, Universitas Dian Nuswantoro, Semarang 50131, Indonesia; abu.salam@dsn.dinus.ac.id (A.S.); junta@dsn.dinus.ac.id (J.Z.)
2  Dinus Research Group for AI in Medical Science (DREAMS), Universitas Dian Nuswantoro, Semarang 50131, Indonesia
3  Department of Health Information Management, Universitas Indonesia Maju, Jakarta 12610, Indonesia; adiwjj@uima.ac.id
*  Correspondence: catur.supriyanto@dsn.dinus.ac.id

**Abstract:** This research paper presents a deep-learning approach to early detection of skin cancer using image augmentation techniques. We introduce a two-stage image augmentation process utilizing geometric augmentation and a generative adversarial network (GAN) to differentiate skin cancer categories. The public HAM10000 dataset was used to test how well the proposed model worked. Various pre-trained convolutional neural network (CNN) models, including Xception, Inceptionv3, Resnet152v2, EfficientnetB7, InceptionresnetV2, and VGG19, were employed. Our approach demonstrates an accuracy of 96.90%, precision of 97.07%, recall of 96.87%, and F1-score of 96.97%, surpassing the performance of other state-of-the-art methods. The paper also discusses the use of Shapley Additive Explanations (SHAP), an interpretable technique for skin cancer diagnosis, which can help clinicians understand the reasoning behind the diagnosis and improve trust in the system. Overall, the proposed method presents a promising approach to automated skin cancer detection that could improve patient outcomes and reduce healthcare costs.

**Keywords:** deep learning; skin cancer; image augmentation; GAN; geometric augmentation; image classification; interpretable technique

## 1. Introduction

Skin cancer is one of the most prevalent and potentially life-threatening forms of cancer worldwide. For more effective therapy and better patient recovery, early detection and diagnosis are essential [1,2]. In recent years, the study of medical image analysis has been completely transformed by convolutional neural networks (CNNs) compared to other advanced machine learning models, supervised or unsupervised, such as k-nearest neighbor (KNN) and support vector machine (SVM), offering a promising approach for the automated detection of skin cancer [3]. CNNs have shown to be extremely effective at extracting complicated patterns and characteristics from medical images, making them an ideal tool for automating the process of skin cancer detection. This technology has the potential to assist dermatologists and healthcare professionals in identifying skin lesions and distinguishing between benign and malignant tumors.

HAM10000 and the International Skin Imaging Collaboration (ISIC) are two datasets that are widely utilized in skin cancer detection studies. HAM10000 is a comprehensive dataset containing diverse dermoscopic images of pigmented skin lesions, a common category of skin cancer [4]. An advantage of the HAM10000 dataset lies in its relatively smaller size compared to the expansive ISIC dataset. This may be beneficial for researchers facing limited computational resources or who want to focus on a specific subset of skin lesions. However, the ISIC dataset has unique advantages, including a larger scale and the inclusion of additional metadata such as lesion location and patient age. The ISIC datasets

have been used for segmentation tasks, but the availability of delineated segmentation masks is limited compared to the classification tasks [5]. The choice of dataset often depends on the specific research investigation and the resources available for the study.

The issue with skin cancer detection datasets is the imbalance in the number of data samples across different classes. This imbalance is observed in both the HAM10000 and ISIC 2017–2020 datasets. In the HAM10000 dataset, which includes a total of 10,015 images, the highest number of data samples can be seen in the melanoma category, with 6705 images, while the lowest number of samples is present in the dermatofibroma category, consisting of 115 images [6]. Meanwhile, in the ISIC 2020 dataset, encompassing a total of 33,126 images, the most abundant data samples are found within the unknown (benign) category, which comprises 27,126 images, whereas the solar lentigo category contains the fewest data samples, with only 7 images [7]. Data imbalance can lead to biased results in classification because the model may be more likely to predict the overrepresented class.

Several approaches can be employed to address imbalances in the amount of data, such as geometric-transformation-based augmentation, feature-space augmentation, and GAN-based augmentation [8]. Geometric data augmentation is a technique employed in the areas of machine learning and computer vision to enhance the variability of a dataset by doing geometric modifications on the original data. The technique transforms the geometric configuration of images by moving the positions of individual pixels without modifying the values of those pixels. These transformations involve altering the position, orientation, or scale of the data while preserving their inherent characteristics. Geometric data augmentation is particularly useful for image data and is often applied to improve the performance of deep learning models. Some common geometric augmentations include rotation, scaling, translation, shearing, flipping, cropping, and zooming.

In feature-space data augmentation, there are two approaches: namely, the undersampling and oversampling approaches. In the undersampling approach, the number of samples from the majority class is reduced to create a more balanced distribution between the classes. By reducing the number of majority class samples, undersampling can help prevent the model from being biased towards the majority class and can improve its ability to recognize the minority class. However, undersampling may result in a loss of potentially valuable information, so it should be applied carefully. In the oversampling approach, additional samples from the minority class are generated to create a more balanced distribution between the classes. The goal is to increase the representation of the minority class to match the number of samples in the majority class, making the dataset more balanced. There are several oversampling methods, with one of the most commonly used techniques being SMOTE (Synthetic Minority Over-sampling Technique). In order to generate synthetic samples for the minority class, SMOTE [9] interpolates between existing data points. This helps to improve the model's ability to learn from the minority class and can lead to better classification results. However, it is important to be cautious with oversampling, as generating too many synthetic samples can lead to overfitting and reduced model generalization.

The concept of GAN-based augmentation refers to the use of generative adversarial networks (GANs) for the purpose of producing synthetic data samples that can be used to augment an existing dataset [10]. This technique is particularly useful in cases where the original dataset is small or imbalanced, as it can help to increase the size of the dataset and balance the class distribution. Augmentation by GAN has been implemented effectively in numerous domains, including medical imaging, natural language processing, and computer vision. Some examples of GAN augmentation in medical imaging include the generation of synthetic CT scans, MRI images, and X-ray images to aid in disease diagnosis and treatment.

In general, resampling approaches can be divided into two categories: namely, input-space data augmentation and feature-space data augmentation. Input-space resampling involves manipulating the original data instances themselves before any feature extraction. Meanwhile, feature-space resampling is applied after feature extraction. Geometric transformations and GAN-based augmentations are categorized as input-space data augmentation,

whereas SMOTE is classified as feature-space data augmentation. The benefit of input-space data augmentation is its independence from the feature extraction method, providing greater flexibility in choosing feature extraction methods. Therefore, in this study, we propose a two-step augmentation, including geometric and GAN-based augmentation, for early detection of skin cancer. The main contributions of this research article are:

- The integration of geometric and GAN-based augmentation for skin cancer detection;
- In this study, we provide an explainable AI using SHAP to explain how the model makes decisions or predictions.

## 2. Related Works

The related studies in this research are categorized into three groups: studies using feature-space augmentation, geometric augmentation, and GAN-based augmentation. Augmentation or oversampling is employed to address the issues of limited data and imbalanced data. Both of these problems contribute to the reduced accuracy of the detection model. This is also observed in skin cancer detection. Several studies have been conducted regarding the use of augmentation or oversampling in skin cancer detection. Abayomi et al. [11] proposed a data augmentation strategy that entails creating a new skin melanoma dataset using dermoscopic images from the publicly available PH2 dataset. The study adopted SMOTE-conv [12], which is a variant of SMOTE. SMOTE-conv utilizes a covariance matrix to detect relationships among attributes and generate synthetic instances. The SqueezeNet deep learning network was then trained using these modified images. In the binary classification scenario, it resulted in an accuracy of 92.18%, while in the multiclass classification scenario, it achieved an accuracy of 89.2%.

SMOTE is an oversampling method used to balance the number of samples between the majority and minority classes in a dataset. SMOTE randomly selects samples from the minority class and creates new synthetic samples by combining them with those of their nearest neighbors. This helps improve the classification performance on imbalanced datasets. However, SMOTE tends to introduce noise and affect classification performance. Therefore, K-means-SMOTE [13] was developed to address SMOTE's limitations. It does this by using k-means clustering to group samples and generate synthetic samples only within clusters with fewer minority class instances. Chang et al. [14] adopted Kmeans-SMOTE to address class imbalance in the ISIC 2018 and ISIC 2019 datasets. Five pre-trained models—namely, VGG16, MELA-CNN, InceptionResNetV2, Inception V3, and the dermatologist handcrafted method—were used to extract features. The minority class data are oversampled using Kmeans-SMOTE and then classified using the Extreme Gradient Boosting (XGB) classifier. The research yielded an accuracy of 96.5%, precision of 97.4%, recall of 87.8%, AUC (Area Under the Curve) of 98.1%, and F1-score of 90.5%.

A study on a deep-learning-based skin cancer classification network (DSCC_Net) was proposed by Tahir et al. [15]; the study proposes the development of a deep learning model with multi-classification capabilities for the purpose of identifying skin cancer through the analysis of dermoscopic pictures. The model was trained and evaluated on three public datasets (HAM10000, ISIC2020, and DermIS), and the results showed that DSCC_Net outperformed other state-of-the-art models in terms of accuracy, sensitivity, specificity, and F1-score. The SMOTE Tomek [16] technique is used to balance the dataset by generating synthetic samples for the minority class and removing noisy and borderline examples from both the minority and majority classes. The DSCC_Net model demonstrates a notable level of performance, achieving an accuracy rate of 94.17%, a recall rate of 93.76%, an F1-score of 93.93%, a precision rate of 94.28%, and an AUC of 99.42%.

An alternative approach is demonstrated by Alam et al. [17], who proposed geometric augmentation in skin cancer detection. Data augmentation involved cropping the images to $256 \times 256$, horizontal flipping, and rotation at various angles. The study utilized the HAM10000 dataset, which initially consisted of 10,015 samples but which was increased to over 30,000 images through data augmentation. Feature extraction was performed using AlexNet, InceptionV3, and RegNetY-320. The proposed method achieved accuracy, F1, and

ROC values of 91%, 88.1%, and 0.95, respectively. A similar approach was also carried out by Sae Lim et al. [18], who proposed geometric augmentation techniques, including rotation, zooming, shifting, and flipping. Experiments were performed using MobileNet on the HAM10000 dataset, leading to performance metrics of accuracy 83.23%, specificity 87%, sensitivity 85%, and an F1-score of 82%.

Alsaidi et al. [19] demonstrated various augmentation techniques in skin cancer detection. Their research proposed the use of GAN to address imbalanced data. Several pre-trained models, including EfficientNet-B0, ResNet50, ViT, and ConvNeXT, were employed. The utilization of GAN as augmentation and EfficientNet-B0 on the HAM1000 dataset yielded an accuracy rate of 96.8%, precision rate of 96.8%, recall rate of 96.9%, and F1-score of 96.8%. The development of GAN models for data augmentation was conducted by Qin et al. [20], who proposed style-based GANs. This method was tested on the ISIC 2018 dataset and achieved an accuracy of 95.2%. Using the same dataset, Ali et al. [21] proposed progressive generative adversarial networks (PGANs) and achieved an accuracy of 70.1%.

## 3. Materials and Methods

### 3.1. Dataset

HAM10000 (https://dataverse.harvard.edu/dataset.xhtml?persistentId=doi:10.7910/DVN/DBW86T, accessed on 2 July 2023) is a dataset containing clinical images of various pigmented skin lesions, including both malignant (cancerous) and benign cases. The dataset consists of 10,015 dermatoscopic images of various skin lesions. These images vary in their types and characteristics. The data are categorized into seven categories based on the type of skin lesion. These categories include melanocytic nevi (*nv*), melanoma (*mel*), benign-keratosis-like lesions (*bkl*), basal cell carcinoma (*bcc*), actinic keratoses and intraepithelial carcinoma (*akiec*), vascular lesions (*vasc*), and dermatofibroma (*df*). Figures 1 and 2 show the number and image samples of each category, respectively. Every image in the collection is accompanied by clinical metadata that includes information such as patient age and gender and the location of the skin lesion. Dermatology experts have provided annotations and diagnoses for each image in this dataset. These annotations include information about the type of lesion (whether it is malignant or benign) and its characteristics. The images in the HAM10000 dataset are of high resolution and good quality, making them suitable for in-depth analysis and diagnosis. HAM10000 is widely used by researchers and machine learning practitioners to develop and evaluate algorithms for skin cancer diagnosis. These data have played a crucial role in advancing the field of computer-aided skin cancer diagnosis. HAM10000 is a publicly available dataset, allowing researchers and developers to access and use it for non-commercial purposes.

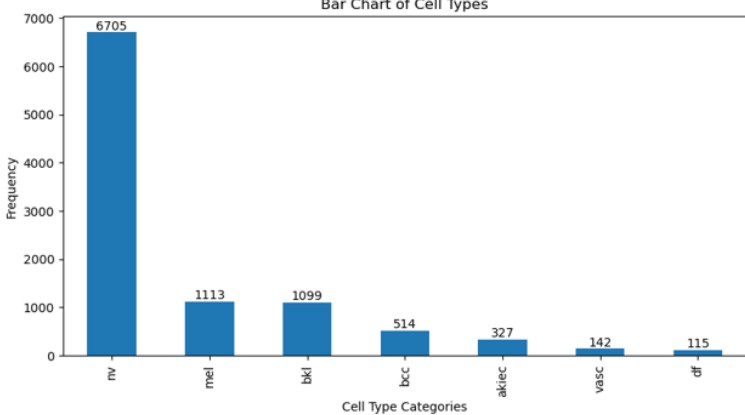

**Figure 1.** Category distribution of HAM10000 dataset.

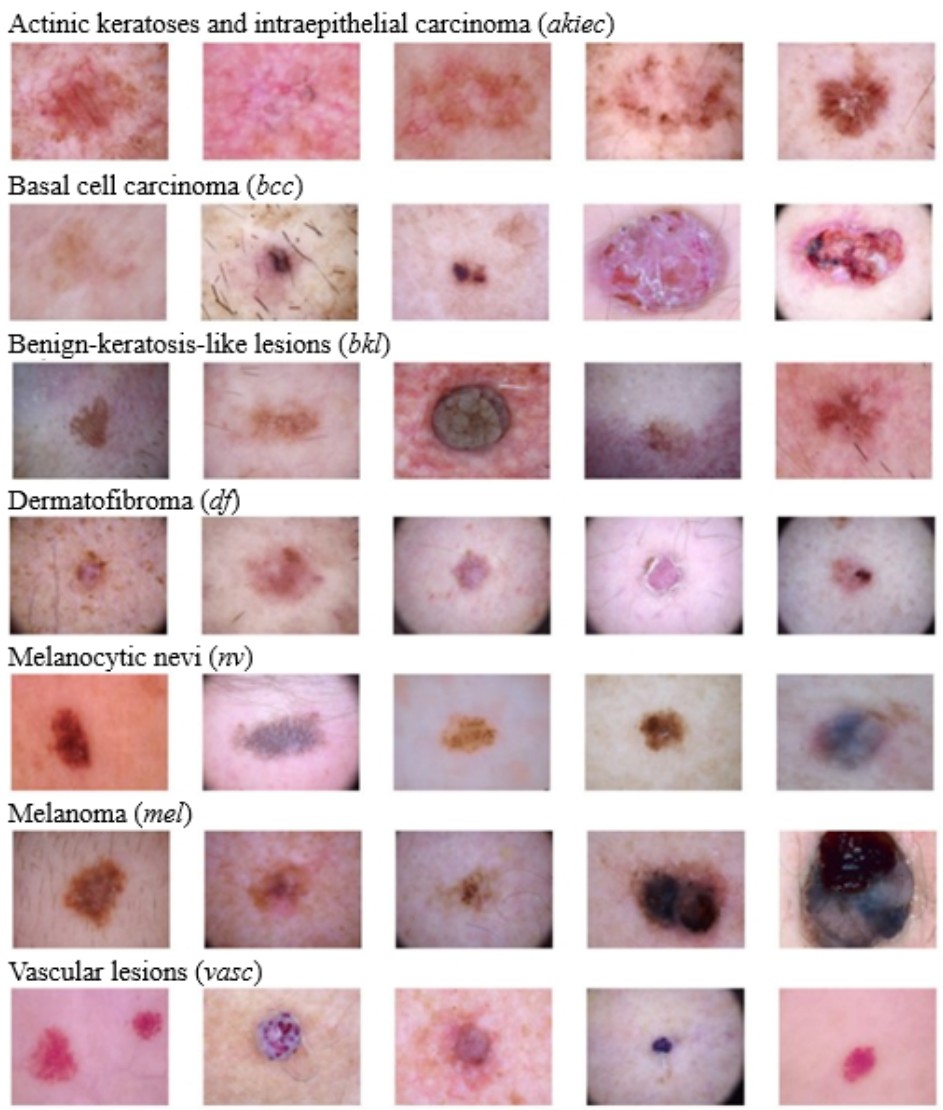

**Figure 2.** Example image of each class in HAM10000 dataset. From top to bottom: *akiec*, *bcc*, *bkl*, *df*, *nv*, *mel*, and *vasc*.

### 3.2. The Proposed Skin Cancer Detection Method

Transfer learning is especially beneficial when there is minimal data for the new task or when building a deep model from scratch would be computationally expensive and time-consuming. In this study, skin cancer classification employs six pre-trained CNN models, which include Xception, Inceptionv3, Resnet152v2, EfficientnetB7, InceptionresnetV2, and VGG19. In order to build a robust model, we apply augmentation techniques to categories that have a limited number of images. Two-stage input-space augmentations—namely, geometric and GAN augmentations—are proposed. Figure 3 shows the flow of skin cancer detection with the proposed augmentation.

Geometric augmentation is one of the data augmentation techniques used in computer image processing, particularly in the context of deep learning and pattern recognition. The goal of geometric augmentation is to enhance the diversity of training data by altering the geometry of the original image without changing the associated labels or class information related to that image. In this way, machine learning models can learn more general patterns and are not overly dependent on specific poses, orientations, or geometric transformations.

Some commonly used geometric augmentation techniques in deep learning include:

1. Rotation: images can be rotated by a certain angle, either clockwise or counterclockwise.
2. Translation: images can be shifted in various directions, both horizontally and vertically.
3. Scaling: images can be resized to become larger or smaller.
4. Shearing: images can undergo linear distortions, such as changing the angles.
5. Flipping: images can be flipped horizontally or vertically.
6. Cropping: parts of the image can be cut out to create variations.
7. Perspective Distortion: images can undergo perspective distortions to change the viewpoint.

By applying these geometric augmentation techniques, training data can be enriched with geometric variations, which helps machine learning models become more robust to variations in real-world images. This allows the model to perform better in pattern recognition tasks, such as object classification, object detection, or image segmentation, even when objects appear in different orientations or poses.

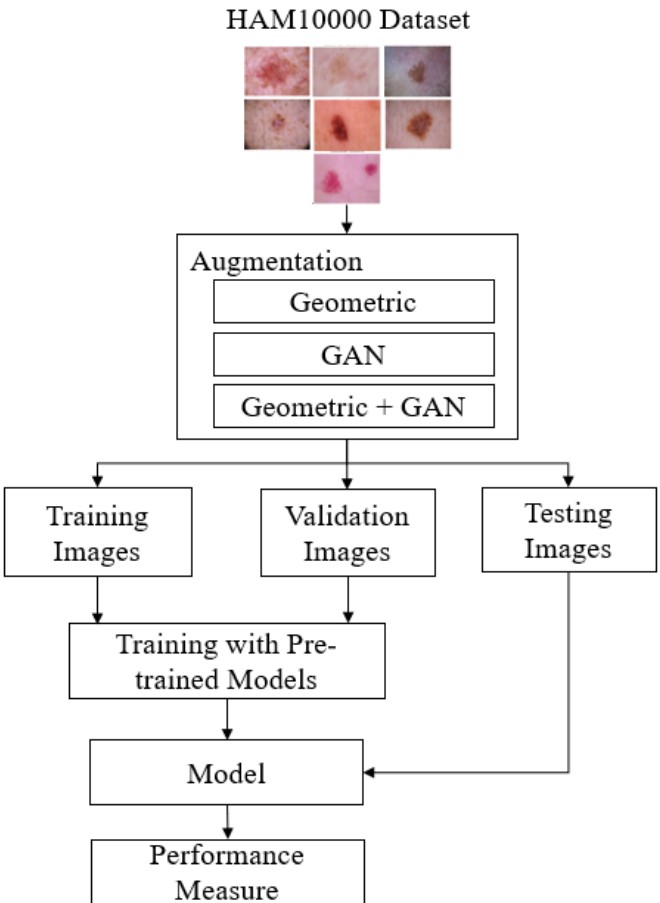

**Figure 3.** The proposed skin cancer detection method.

GAN [22] augmentation refers to the use of generative adversarial networks (GANs) as one of the data augmentation techniques in the context of machine learning, especially in image processing. GAN is an artificial neural network architecture consisting of two models, the generator and the discriminator, that compete in a game to improve their capabilities [23]. In the context of data augmentation, GAN augmentation involves using the GAN generator to create additional data that are similar to the existing training data. The GAN generator tries to create images that appear authentic, while the GAN discriminator attempts to distinguish between images generated by the generator and real images.

By combining the images generated by the GAN generator with the training data, the dataset can be enhanced with image variations that appear realistic. GAN augmentation has been proven effective at improving the performance of machine learning models, especially in image recognition tasks such as object classification, object detection, or image segmentation, as it can create more diverse and relevant image variations.

### 3.3. Design of Experiments

In the experiment, 20% of the 10,015 images, which are 2003 images, are utilized for testing, while the remaining 8012 images are split into 90% (7210) for training and 10% (802) for validation. Three methods are used to oversample the data: geometric, GAN, and geometric+GAN augmentations. Experiments are carried out using Python 3.11.5 and were run on an Nvidia DGX Station A100 with a 40 GB GPU, a 64-core CPU, and 512 GB of DDR4 RAM.

Several experimental schemes are established to achieve the best performance. In the first scheme, skin cancer detection is conducted using the original data (without augmentation). In the second, third, and fourth schemes, the original data are augmented using geometric augmentation, GAN augmentation, and geometric+GAN augmentation, respectively. This study uses rotation, shift, shear, zoom, flip, and brightness for geometric augmentation, with detailed parameter values shown in Table 1. During GAN-based augmentation, a total of 1000 epochs are run with a batch size of 64. Table 2 shows the structure of discriminator and generator networks of GAN-based augmentation. In the discriminator, we employ the Adam optimizer with a learning rate of 0.0002 along with binary cross-entropy as the loss function. LeakyReLU with $\alpha = 0.2$ is applied as the activation function for all layers except the last one, where a sigmoid activation function is utilized. A discriminator dropout with a probability of 0.2 is applied. Also in the generator network, each layer utilizes LeakyReLU with $\alpha = 0.2$ except for the final layer, which employs Tanh as the activation function. The parameter values of the training model, such as optimizer, learning rate, and epoch, are shown in Table 3. In this experiment, we also conduct trials with a custom FC layer configuration as shown in Table 4, consisting of a dense layer with 64 neurons, a dense layer with 32 neurons, and a dense layer with 7 neurons [24].

**Table 1.** Summary of geometric augmentation parameters.

| Parameter | Value |
|---|---|
| rotation_range | 20 |
| width_shift_range | 0.2 |
| height_shift_range | 0.2 |
| shear_range | 0.2 |
| zoom_range | 0.2 |
| horizontal_flip | True |
| brightness_range | (0.8, 1.2) |

This study also uses SHAP to explain skin cancer detection, which is a technique or approach that utilizes the concept of Shapley values to explain the contribution of each pixel or feature in an image to the model's predictions. CNNs are frequently referred to as black boxes due to the difficulty in deciphering their decision-making processes. For the purpose of understanding model behavior and building trust, SHAP assists in improving the transparency and interpretability of the CNN's decision-making. In SHAP, the concept of Shapley values is applied to measure and understand the influence of each pixel in the image on the model's output or prediction. This technique is valuable for interpreting machine learning models, including the convolutional neural network (CNN) models frequently used for image-based tasks. Positive SHAP values signify that the presence of a pixel had a positive impact on the prediction (red pixel), whereas negative values indicate the contrary (blue pixel) [25].

**Table 2.** Summary of GAN-based augmentation parameters.

|  | Layer | Activation |
| --- | --- | --- |
| Discriminator | Conv2D | LeakyReLU |
|  | Conv2D | LeakyReLU |
|  | Conv2D | LeakyReLU |
|  | Conv2D | LeakyReLU |
|  | Flatten |  |
|  | Dropout |  |
|  | Dense | Sigmoid |
| Generator | Dense | LeakyReLU |
|  | Conv2DTranspose | LeakyReLU |
|  | Conv2DTranspose | LeakyReLU |
|  | Conv2DTranspose | LeakyReLU |
|  | Conv2D | Tanh |

**Table 3.** Parameters of training model.

| Parameter | Value |
| --- | --- |
| Optimizer | Adam |
| Learning rate | 0.0001 |
| Optimizer parameters | beta_1 = 0.9, beta_2 = 0.999 |
| Epochs | 100 (with early stopping) |

**Table 4.** Summary of custom FC layers.

| Layer | Output Shape | Activation |
| --- | --- | --- |
| Dense | (None, 64) | Relu |
| Dense | (None, 32) | Relu |
| Dense | (None, 7) | Softmax |

*3.4. Performance Metrics*

The evaluation of performance was conducted using seven metrics: accuracy (Acc), precision (Prec), recall (Rec), F1-score, SpecificityAtSensitivity, SensitivityAtSpecificity, and G-mean. Accuracy assesses the proportion of true positives and true negatives among all the images. Precision is a metric that quantifies the accuracy of a model's positive predictions. It is calculated by dividing the number of accurate positive predictions by the total number of positive predictions. Recall, also known as sensitivity, measures the ratio of true positives to all relevant elements, i.e., the true positives in the dataset. Specificity is a metric that assesses the model's capability to accurately recognize instances that are actually not part of the positive class in a classification scenario. The F1-score represents the harmonic mean of recall and precision, providing an indication of classification accuracy in imbalanced datasets. Equations (1)–(6) define these seven metrics. G-mean, short for geometric mean, is utilized to assess the effectiveness of classification models, particularly in situations where imbalanced datasets exist.

$$Accuracy = \frac{TP + TN}{TP + TN + FP + FN} \tag{1}$$

$$Precision = \frac{TP}{TP + FP} \tag{2}$$

$$Recall = Sensitivity = \frac{TP}{TP + FN} \tag{3}$$

$$F1\text{-}score = \frac{2 * Precision * Recall}{Precision + Recall} \tag{4}$$

$$Specificity = \frac{TN}{TN + FP} \tag{5}$$

$$\textit{G-mean} = \sqrt{sensitivity \times specificity} \tag{6}$$

## 4. Results and Discussion

Before the prediction process, data augmentation is performed on the training, validation, and testing data in the HAM10000 dataset using geometric augmentation, GAN augmentation, and geometric+GAN augmentation. The limited number of images in the skin cancer class is augmented to bring it closer to the number of images in the class with the highest number of images (*nv* class). The number of images in each class before and after augmentation is shown in Table 5.

Table 6 shows the performance comparison of several pre-trained models with the proposed augmentation method. Using the original data, Resnet152v2 performed the best based on accuracy (84.12%), precision (84.77%), recall (83.67%), and F1-score (84.22%). However, when considering sensitivity, specificity, and G-mean, EfficientnetB7 achieved the best metric values with 99.49%, 94.91%, and 97.17%, respectively. Through the augmentation scheme we proposed, the accuracy of skin cancer detection can be enhanced, reaching a range of 96% to 97.95%. Overall, geometric augmentation produced the best performance based on accuracy, precision, and F1-score metrics, while geometric+GAN yielded the best metrics in terms of sensitivity, specificity, and G-mean values. SensitivityAtSpecificity, SpecificityAtSensitivity, and G-mean all approach 100% when employing geometric+GAN on a tested pre-trained model. It is clear from Table 7 that changing the FC layer makes the accuracy go up to 98.07% when EfficientnetB7 and geometric augmentation are used.

**Table 5.** Distribution of each skin cancer category for each augmentation scheme.

| Category | Original | | | Geometric Aug. | | | GAN | | | Geometric Aug.+GAN | | |
|---|---|---|---|---|---|---|---|---|---|---|---|---|
| | Train | Test | Val | Train | Test | Val | Train | Test | Val | Train | Test | Val |
| **vasc** | 110 | 26 | 6 | 4801 | 1350 | 554 | 4843 | 1359 | 503 | 4805 | 1371 | 529 |
| **nv** | 4822 | 1347 | 536 | 4826 | 1316 | 563 | 4854 | 1302 | 549 | 4856 | 1300 | 549 |
| **mel** | 792 | 222 | 99 | 4877 | 1303 | 525 | 4858 | 1319 | 528 | 4836 | 1361 | 508 |
| **df** | 83 | 25 | 7 | 4775 | 1423 | 507 | 4831 | 1325 | 549 | 4831 | 1340 | 534 |
| **bkl** | 785 | 224 | 90 | 4887 | 1316 | 502 | 4813 | 1329 | 563 | 4831 | 1337 | 537 |
| **bcc** | 370 | 101 | 43 | 4783 | 1360 | 562 | 4792 | 1387 | 526 | 4798 | 1394 | 513 |
| **akiec** | 248 | 58 | 21 | 4844 | 1319 | 542 | 4802 | 1366 | 537 | 4836 | 1284 | 585 |
| **Num. images** | 7210 | 2003 | 802 | 33,793 | 9387 | 3755 | 33,793 | 9387 | 3755 | 33,793 | 9387 | 3755 |
| **Total images** | 10,015 | | | 46,935 | | | 46,935 | | | 46,935 | | |

Figure 4 shows sample accuracy results from the training and validation of EfficientnetB7 on the original dataset and the proposed augmentation. The training and validation accuracies appear to overfit the original dataset (Figure 4a). Validation accuracy is improved by geometric augmentation (Figure 4b), thereby reducing overfitting. Training accuracy is enhanced through the use of GAN and geometric+GAN (Figure 4c,d).

The sample confusion matrices generated from the original dataset and the best-proposed model are shown in Figures 5 and 6, respectively. Both of these confusion matrices were generated using EfficientnetB7. In Figure 5, many classes are still predicted inaccurately due to imbalanced data. In Figure 6, skin cancer images in the *df* and *vasc* classes can be accurately classified with no classification errors. Only 3 images out of 1319 images in the *akiec* class were misclassified as *bcc*. Six mispredictions were observed among *bcc* samples out of 1360. Fifty-two instances of *bkl* samples were inaccurately predicted out of a comprehensive pool of 1316 samples. Out of the overall 1303 samples,

70 samples belonging to the *mel* class were predicted incorrectly. Similarly, for *nv* cases, 50 mistakes were found in 1316 samples.

**Table 6.** Performance of the proposed augmentation method on several pre-trained models.

| Augmentation Method | Pre-Trained Model | Acc | Prec | Rec | F1 | Sensitivity AtSpecificity | Specificity AtSensitivity | G-Mean | Epoch |
|---|---|---|---|---|---|---|---|---|---|
| Original Data | Xception | 79.93 | 80.70 | 79.53 | 80.11 | 99.16 | 91.71 | 95.36 | 12 |
| | Inceptionv3 | 78.88 | 79.51 | 78.48 | 78.99 | 99.31 | 92.31 | 95.75 | 11 |
| | Resnet152v2 | 84.12 | 84.77 | 83.67 | 84.22 | 99.33 | 93.56 | 96.40 | 18 |
| | EfficientnetB7 | 78.03 | 79.63 | 77.28 | 78.44 | 99.49 | 94.91 | 97.17 | 11 |
| | InceptionresnetV2 | 79.63 | 80.20 | 78.68 | 79.44 | 99.28 | 91.76 | 95.45 | 19 |
| | VGG19 | 81.73 | 81.83 | 81.63 | 81.73 | 99.12 | 91.26 | 95.11 | 28 |
| Geometric | Xception | 97.05 | 97.06 | 97.01 | 97.03 | 99.87 | 99.12 | 99.49 | 19 |
| | Inceptionv3 | 97.38 | 97.48 | 97.35 | 97.41 | 99.90 | 99.20 | 99.55 | 31 |
| | Resnet152v2 | 96.90 | 96.95 | 96.86 | 96.90 | 99.85 | 98.93 | 99.39 | 28 |
| | EfficientnetB7 | 97.95 | 98.00 | 97.90 | 97.95 | 99.91 | 99.41 | 99.66 | 19 |
| | InceptionresnetV2 | 97.40 | 97.46 | 97.36 | 97.41 | 99.89 | 99.20 | 99.55 | 28 |
| | VGG19 | 97.22 | 97.24 | 97.20 | 97.22 | 99.83 | 98.84 | 99.33 | 32 |
| GAN | Xception | 96.08 | 96.35 | 95.96 | 96.16 | 99.86 | 98.70 | 99.28 | 10 |
| | Inceptionv3 | 96.50 | 96.62 | 96.45 | 96.53 | 99.86 | 98.64 | 99.25 | 16 |
| | Resnet152v2 | 96.30 | 96.47 | 96.23 | 96.35 | 99.79 | 98.37 | 99.08 | 20 |
| | EfficientnetB7 | 96.48 | 96.59 | 96.44 | 96.51 | 99.79 | 98.25 | 99.02 | 20 |
| | InceptionresnetV2 | 96.22 | 96.32 | 96.20 | 96.26 | 99.82 | 98.44 | 99.13 | 18 |
| | VGG19 | 96.22 | 96.26 | 96.20 | 96.23 | 99.70 | 100.00 | 99.85 | 38 |
| Geometric + GAN | Xception | 96.21 | 96.51 | 96.04 | 96.27 | 99.94 | 99.22 | 99.58 | 9 |
| | Inceptionv3 | 96.45 | 96.56 | 96.39 | 96.48 | 99.86 | 98.56 | 99.21 | 20 |
| | Resnet152v2 | 96.59 | 96.75 | 96.45 | 96.60 | 99.86 | 98.87 | 99.36 | 14 |
| | EfficientnetB7 | 96.50 | 96.61 | 96.43 | 96.52 | 99.89 | 98.92 | 99.40 | 14 |
| | InceptionresnetV2 | 96.71 | 96.82 | 96.67 | 96.74 | 99.85 | 98.62 | 99.23 | 21 |
| | VGG19 | 95.39 | 97.36 | 93.89 | 95.59 | 100.00 | 99.93 | 99.96 | 17 |

**Table 7.** Performance of the proposed augmentation method on the custom FC layer (three dense layers with 64 neurons, 32 neurons, and 7 neurons, respectively).

| Augmentation Method | Pre-Trained Model | Acc | Prec | Rec | F1 | Sensitivity AtSpecificity | Specificity AtSensitivity | G-Mean | Epoch |
|---|---|---|---|---|---|---|---|---|---|
| Geometric | EfficientnetB7 | 98.07 | 98.10 | 98.06 | 98.08 | 99.92 | 99.46 | 99.69 | 20 |
| GAN | Inceptionv3 | 96.48 | 96.63 | 96.44 | 96.53 | 99.83 | 98.54 | 99.18 | 17 |
| Geometric + GAN | InceptionresnetV2 | 96.90 | 97.07 | 96.87 | 96.97 | 99.86 | 98.90 | 99.38 | 22 |

We performed a comparative analysis to evaluate the performance of our model by comparing it to the outcomes of earlier research that utilized the use of the HAM10000 dataset, as shown in Table 8. Our proposed approach outperforms earlier findings in a number of metrics. Our limitation is mainly in terms of accuracy when compared to Gomathi et al. [4]. The accuracy rate still needs improvement, and we plan to explore other deep-learning architectures to enhance skin cancer detection. However, in terms of recall, precision, and F1, our approach outperforms the previous research. The standard deviations of accuracy, precision, and recall in our proposed methods also indicate low values, suggesting that our proposed approach demonstrates consistent performance across all three metrics. Figure 7 shows the SHAP explanations of *akiec*, *bcc*, *bkl*, *df*, *mel*, *nv*, and *vasc* samples. The explanations are displayed on a clear grey background, with the testing images on the left.

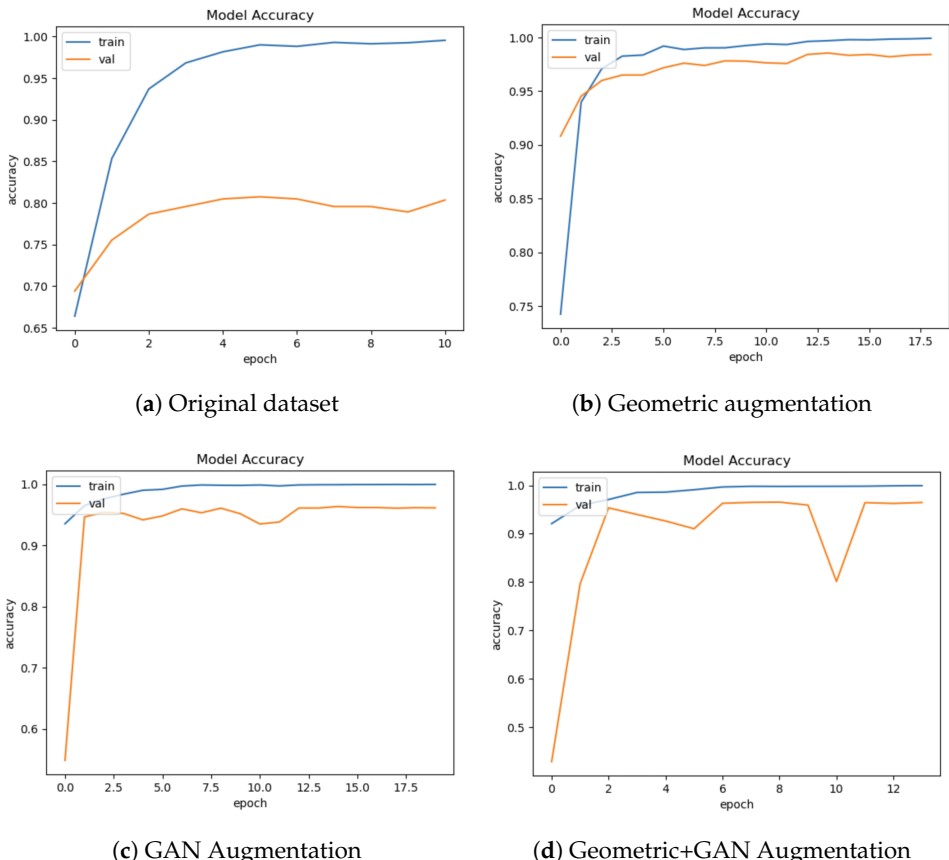

(**a**) Original dataset         (**b**) Geometric augmentation

(**c**) GAN Augmentation         (**d**) Geometric+GAN Augmentation

**Figure 4.** The samples of training and validation accuracy on EfficientnetB7.

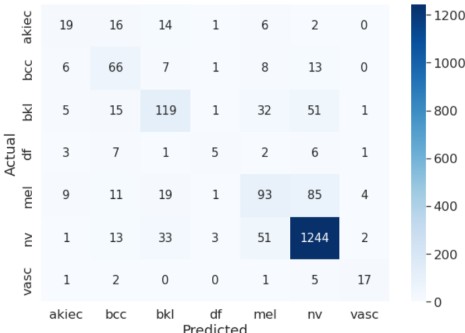

**Figure 5.** Confusion matrix of EfficientnetB7 on original dataset.

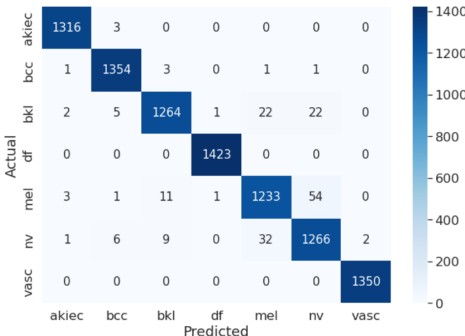

**Figure 6.** Confusion matrix of the best performance model (EfficientnetB7+Custom FC using geometric augmentation).

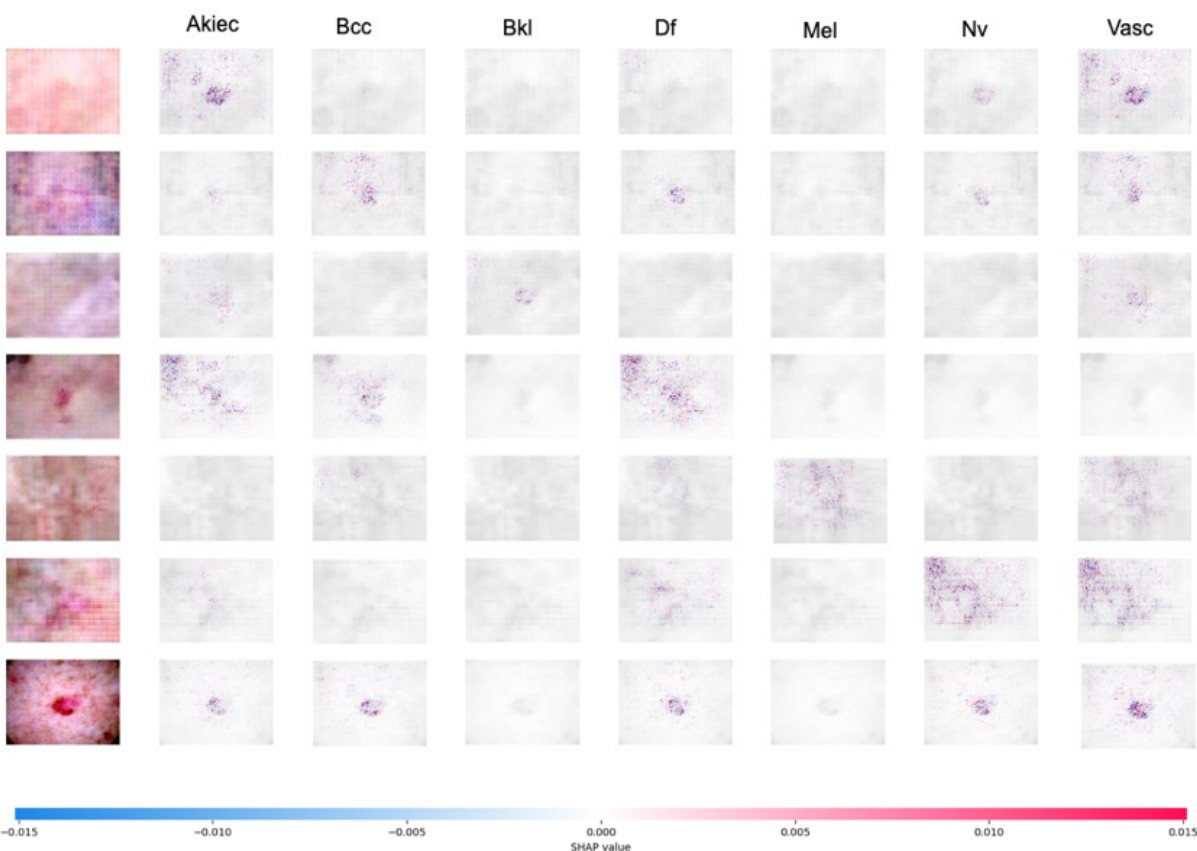

**Figure 7.** The results of SHAP explanation on InceptionResnetV2 using Geometric+GAN augmentation. The sample images are correctly classified as *akiec*, *bcc*, *bkl*, *df*, *mel*, *nv*, and *vasc* since the high concentrations of red pixels are located in the second, third, fourth, fifth, sixth, seventh, and eighth explanation column images, respectively.

**Table 8.** A comparative analysis of performance with the latest models.

| Ref. | Method | Acc | Prec | Rec | F1 | Stdev |
|---|---|---|---|---|---|---|
| Alam et al. [17] | AlexNet, InceptionV3, and RegNetY-320 | 91 | - | - | 88.1 | - |
| Kalpana et al. [2] | ESVMKRF-HEAO | 97.4 | 96.3 | 95.9 | 97.4 | 0.7767 |
| Shan et al. [26] | AttDenseNet-121 | 98 | 91.8 | 85.4 | 85.6 | 6.3003 |
| Gomathi et al. [4] | DODL net | 98.76 | 96.02 | 95.37 | 94.32 | 1.7992 |
| Alwakid et al. [27] | InceptionResnet-V2 | 91.26 | 91 | 91 | 91 | 0.1501 |
| Sae-Lim et al. [18] | Modified MobileNet | 83.23 | - | 85 | 82 | - |
| Ameri [28] | AlexNet | 84 | - | - | - | - |
| Chaturvedi et al. [6] | ResNeXt101 | 93.2 | 88 | 88 | - | 3.0022 |
| Shahin Ali et al. [29] | DCNN | 91.43 | 96.57 | 93.66 | 95.09 | 2.5775 |
| Sevli et al. [30] | Custom CNN architecture | 91.51 | - | - | - | - |
| Fraiwan et al. [31] | DenseNet201 | 82.9 | 78.5 | 73.6 | 74.4 | 4.6522 |
| Balambigai et al. [32] | Grid search ensemble | 77.17 | - | - | - | - |
| Shaheen et al. [33] | PSOCNN | 97.82 | - | - | 98 | - |
| This study | Geometric+EfficientnetB7+Custom FC | 98.07 | 98.10 | 98.06 | 98.08 | 0.0002 |
|  | GAN+InceptionV3 | 96.50 | 96.62 | 96.45 | 96.53 | 0.0009 |
|  | Geometric+GAN+InceptionresnetV2+Custom FC | 96.90 | 97.07 | 96.87 | 96.97 | 0.0011 |

## 5. Conclusions

This study provides valuable insights into a deep-learning approach for the early detection of skin cancer using image augmentation techniques. The proposed two-stage image augmentation technique, involving both geometric augmentation and GAN augmentation, demonstrated high performance. The proposed model achieves an accuracy of

96.90%, precision of 97.07%, recall of 96.87%, and F1-score of 96.97%. The other metrics, such as sensitivity, specificity, and G-mean, of the proposed augmentation method also achieve better performance compared to the results from the original dataset. The use of an interpretable technique for skin cancer diagnosis is also a significant contribution to the field, as it can help clinicians understand the reasoning behind the diagnosis and improve trust in the system. Overall, this research paper presents a promising approach to automated skin cancer detection that could have a significant impact on patient outcomes and healthcare costs. For future research, we will include another dataset, namely ISIC 2020, to validate the results of the next experiments.

**Author Contributions:** Conceptualization, C.S.; Data curation, A.S. and J.Z.; Formal analysis, A.W.; Investigation, C.S. and A.W.; Methodology, C.S.; Resources, A.S.; Software, A.S. and J.Z.; Supervision, A.W.; Validation, C.S. and A.W.; Writing—original draft, C.S.; Writing—review and editing, C.S. All authors have read and agreed to the published version of the manuscript.

**Funding:** This research was funded by DRTPM-DIKTI: 065/A38-04/UDN-09/VII/2023 for research funding in 2023.

**Data Availability Statement:** Dataset is publicly available at: https://dataverse.harvard.edu/dataset.xhtml?persistentId=doi:10.7910/DVN/DBW86T (accessed on 2 July 2023)

**Conflicts of Interest:** The authors declare no conflict of interest.

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
