# Peer review of "Two-Stage Input-Space Image Augmentation and Interpretable Technique for Accurate and Explainable Skin Cancer Diagnosis"

_computation, doi:10.3390/computation11120246_

Round 1

Reviewer 1 Report

Comments and Suggestions for Authors

It is recommended that the authors include a clear and concise statement detailing the key contributions of this work.

To improve readability and maintain consistency, the formatting of figure and table captions should be standardized.

The abstract exhibits grammatical problems, including run-on phrases and punctuation faults, which might impair comprehension. The abstract must be revised for clarity and accuracy.

The value of accuracy received using proposed model must be mentioned in the conclusion which is missing.

Visual aids like heatmaps, ROC curves, PR curves, and confusion matrices make it much easier to understand the data. These pictures do a good job of showing how well different types work. But to have labels and explanations for the numbers to make things more clear is mandatory which is missing.

Comments on the Quality of English Language

It is recommended that the authors include a clear and concise statement detailing the key contributions of this work.

To improve readability and maintain consistency, the formatting of figure and table captions should be standardized.

The abstract exhibits grammatical problems, including run-on phrases and punctuation faults, which might impair comprehension. The abstract must be revised for clarity and accuracy.

The value of accuracy received using proposed model must be mentioned in the conclusion which is missing.

Visual aids like heatmaps, ROC curves, PR curves, and confusion matrices make it much easier to understand the data. These pictures do a good job of showing how well different types work. But to have labels and explanations for the numbers to make things more clear is mandatory which is missing.

Author Response

Thank you very much for your time in reviewing this paper. The reviewer provided excellent suggestions and reminded us of some aspects, such as adding performance values in the conclusion. We hope that our revisions address the questions and suggestions from the reviewer. The corrections we made are highlighted in yellow in the manuscript.

Reviewer 2 Report

Comments and Suggestions for Authors

1. Good abstract

2. why you select space data augmentation and feature space data augmentation? just in your opinion

3. Please rewrite the related works section. please don't write one LR article one paragraph. Please combined and make comprehensive.

4. Put the website/link for database downloaded.

5. Methodology acceptable. 

6. In table 4, please declare the short form for category . What vasc and so on. 

7. Good result. Good comparison with existing study refer by table 7.  

Comments on the Quality of English Language

Make a revision

Author Response

Thank you to the reviewers who provided their valuable suggestions and comments, which have been very beneficial for improving this manuscript. The corrections we made are highlighted in yellow in the manuscript.

Reviewer 3 Report

Comments and Suggestions for Authors

Summary of the Manuscript:

Review for the manuscript titled: "Two-Stages Input Space Image Augmentation and Interpretable Technique for Accurate and Explainable Skin Cancer Diagnosis" by Catur Supriyanto, et, al. The paper presents a deep learning methodology for early detection of skin cancer, utilizing a novel two-stage image augmentation process combining geometric and generative adversarial network (GAN) based augmentations. The study employs several pre-trained convolutional neural networks (CNNs) and interpretable machine learning through SHapley Additive exPlanations (SHAP). Using the HAM10000 dataset, the authors achieve impressive accuracy, precision, recall, and F1-score, outperforming current state-of-the-art methods.

Strengths:

  1. The combination of geometric and GAN-based augmentation is novel and addresses the data imbalance issue effectively.
  2. The model achieves high metrics in accuracy, precision, recall, and F1-score, indicating its robustness.
  3. The use of SHAP for interpretability could help in clinical decision-making, bridging the gap between AI tools and practitioners.

Weaknesses:

  1. The research is based solely on the HAM10000 dataset; additional validation on more datasets would strengthen the findings.
  2. The paper lacks extensive discussion on the model's generalizability to other forms of skin lesions not covered in the dataset.
  3. The use of GANs is computationally intensive; the paper could discuss the trade-offs and feasibility of deployment in practical settings.

Recommendations:

  1. To enhance the robustness and applicability of the model, validation on diverse datasets is recommended.
  2. Some parts of the methodology could be further elaborated for reproducibility, such as the exact GAN architecture and parameters.
  3. The paper would benefit from a discussion on the potential challenges and considerations for deploying such a model in a clinical environment.

Conclusion: The manuscript makes a valuable contribution to the field of medical image analysis for skin cancer detection. The proposed method shows potential for improving early detection and aiding clinicians. With revisions, particularly validation on additional datasets and further discussion on model generalizability and clinical deployment, the paper would be a significant addition to the literature. Therefore, major revision is recommended before publication.

Author Response

Thank you very much for your time in reviewing this paper. The reviewer provided a thorough explanation of the strengths, weaknesses, and offered recommendations and conclusions for this publication. From the explanations, we have noted several aspects that need immediate attention for improvement, namely the need for additional datasets and parameters of the GAN-based augmentation used in the experiments. We hope that our responses below can address the questions and suggestions for improvement from the reviewer. The corrections we made are highlighted in yellow in the manuscript.

Round 2

Reviewer 1 Report

Comments and Suggestions for Authors

1. Fig 7 and table 8 comes in between reference section which is required to be corrected.

Comments on the Quality of English Language

English must be improved. 

Author Response

Thank you for your time and suggestions in reviewing this paper. Thus, the writing in this paper has become better.

Reviewer 3 Report

Comments and Suggestions for Authors

Upon reviewing the authors' response to the reviewer's comments, it is evident that they have comprehensively addressed all concerns, particularly regarding model robustness and methodological clarity. Their commitment to incorporating diverse datasets and detailed explanation of the GAN architecture substantially enhance the manuscript. The authors have demonstrated a commendable level of diligence and responsiveness. Therefore, I recommend the manuscript for publication.

Author Response

(The authors gave the same response as above.)
